# Different Measures of Country Risk: An Application to European Countries

**Guido Bonatti [1], Andrea Ciacci [2,3] and Enrico Ivaldi [3,4,*]**

1   School of Economics and Finance, Queen Mary University of London, London E1 4NS, UK;
    bonattiguido@libero.it
2   Department of Economics—DIEC, University of Genoa, 16126 Genoa, Italy; andrea.ciacci@edu.unige.it
3   Centro de Investigaciones en Econometría—CIE, University of Buenos Aires,
    Buenos Aires C1113 CABA, Argentina
4   Department of Political Science—DISPO, University of Genoa, 16125 Genoa, Italy
*   Correspondence: enrico.ivaldi@unige.it

**Abstract:** Country Risk (CR) is a relevant instrument to analyze and understand economic performances and relationships between different countries in the actual economic and political international globalized context. The present work develops indexes for the European Union countries by applying three different methods in the field of formative approach. Our aim is to show how robust CR measurements can be developed by operational and easily computable methods. We identify a set of significant variables included in the reference literature. Then, we propose three simple aggregative processes in order to obtain CR measures, at a precise time and over time. As a result, if we compare the outcomes, similar CR rankings emerge. In other words, there are no relevant differences in results also due to different methods of applications. The findings demonstrate that the choice of the aggregation method depends on the willingness of the researcher to baste the analysis with or without weighing and, therefore, on the semantic content that is assigned to the entire research structure. Each analysis should follow a disinterested theoretical–methodological consistency, knowing that the choice of a particular indexing process in the field of aggregation does not significantly alter the nature of the results compared to what would result by applying a different method.

**Keywords:** country risk; formative approach; additive indexes



## 1. Introduction

It is a difficult task to deal with the topic of Country Risk (CR) trying to use a unique, clear, and precise definition. The main reason is that, for decades, the literature about the subject has offered a wide range of definitions allowing studies and analysis from different points of view and disciplines. What complicates the search for a univocal framework is the fact that the idea of CR can be defined also by other terms, i.e., "Political Risk", which is used more frequently, and "Cross-border Risk" or "Sovereign Risk" which are utilized less frequently.

During the 1960s, the definition of CR was mainly linked to the exposure of multinational companies to political risk. The 1980s were characterized by debt crises of emerging countries, with the concept of CR acquiring a more economic connotation referring to the solvency assessment of a country (Bekaert et al. 1998), i.e., the risk of default (Estrada 2000). Since the early 1990s, the frequent financial crisis in several countries outlined the third phase. The parties involved are, on the one hand, the insolvent debtor country, on the other hand, the official creditors, i.e., the governments of the industrialized countries and the International Financial Institutions (Baldacci and Chiampo 2007), which play a decisive role in that they are responsible for the provision of funding aimed at macroeconomic stabilization. Since the year 2000, the studies on CR have been expanding more and more using

a range of variables and tools that allowed for a broader definition. The most thorough definition is given by Meldrum (2000), who states that CR is intended as "the set of risks that are not encountered if transactions are done in the domestic market, but which can emerge when an investment in a foreign country is made. These risks are mainly ascribable to the political, economic, and social differences between the country of the investor and the country in which the investment is done". This author also offers a complete analysis of the sources that influence CR, i.e., economic risk, the risk of transfer, the risk deriving from the fluctuation of the exchange rate, the risk of geographical location, the sovereign risk, and the political risk.

It is possible to make a first important distinction between qualitative studies, mainly used between the 1960s and 1970s with the aim of analyzing political risk, and quantitative studies, which started to be considered in the 1980s, with the aim of predicting default situations or financial crises at country level.

In recent decades, the literature has increasingly opened up to a new concept of risk, which is linked to relations with foreign countries. The analysis of CR in the 1970s and 1980s tended to focus on the risk of a private lender (such as a bank) in granting a loan to a sovereign government outside its own country (Meldrum 2000). Today, thanks to globalization and the development of international businesses, markets are increasingly dependent on each other and, therefore, all the world's economies are. Political changes resulting from the fall of communism and the implementation of market-oriented economic and financial reforms resulted in an enormous amount of external capital flowing into the emerging markets of Eastern Europe, Latin America, Asia, and Africa. These events alerted international investors to the fact that the globalization of trade and open capital markets is a risky element that can cause financial crises with rapid contagion effects, threatening the stability of the international financial sector (Platt and Platt 2008). In light of the tumultuous events that have taken place since 11 September 2001, the risks associated with engagement in international relations have increased significantly and have become more difficult to analyze and predict for decision-makers in the economic, financial, and political sectors (Hoti and McAleer 2004). The purpose of the work is to build robust and easily updatable CR indexes. This aim is important for some reasons: at first, we recognize the need to gather the most important and recent advances in the study of the determinants of CR; therefore, we propose the use of indexes of simple construction, whose advantage is to guarantee an agile use and a simple and intuitive reading of the results obtained. Among the quantitative aggregative methods adopted, two are weighted indexes. Finally, we show how the choice of an aggregative method must follow a continuity with respect to the conceptual framework of reference, ensuring continuity along the entire theoretical–methodological aspect. However, the results obtained with the different methods are almost equal. The choice of an aggregation method by the researcher is not so compromising but functional to the analysis of CR.

After the introduction, which offers a description of the concept and the determinants of CR, the second section reviews the recent literature focused in particular on the introduction of new variables and new methodologies of analysis deriving from the opportunities and the problems connected with technological innovations (i.e., Big Data). In the third section, we describe the statistical methodologies applied: the selection of the variables and the different aggregative methods used for constructing CR indexes. In the fourth section, we present the results of several methodologies and their representation, while in the last two paragraphs, we discuss the results and draw the conclusions.

## 2. Analysis of the Literature

When analyzing the literature, the term CR appears to be often traced to financial risk and linked to cross-border investments (Nordal 2001). It is usually analyzed from the point of view of foreign investors and represents their choice to invest in a particular country rather than in another. However, there is a specification that Meldrum offers about the foreign investment risks: in his study, he reveals that when commercial transactions

occur across national borders, there are additional risks not present in domestic transactions. These additional risks typically include risks arising from national differences in economic and political structures, in socio-political, geographic, and currency institutions (Meldrum 2000).

The national differences are multiple and concern different areas, so that CR cannot be analyzed following a single line of research (Doumpos and Zopounidis 2002). A possible division of CR refers to economic risk, commercial risk, and political risk (Nordal 2001). Economic risk, linked to a country's macroeconomics and its development, can be linked to the level of interest rates and to the exchange rate, which can affect the profitability of an investment. Commercial risk concerns the risk of a specific investment, such as the risk related to the fulfillment of contracts with private companies or local partners. Political risk is the most important risk category: each country is a political entity and, as such, it has its own laws and regulations to apply to investments, such as laws protecting property rights. Political risk primarily concerns the possibility of modifying these laws and regulations and the resulting uncertainty is the greatest source of political risk in a country, which affects its reliability. This type of risk can also be caused by the behavior of the State or state companies on the market, or by more extreme situations such as civil unrest or war (Meldrum 2000).

There are three types of political risk (Nordal 2001): the first is the risk of transfer, which occurs when goods and services are "transferred" beyond national borders; the second is operational risk, linked to the risk in functioning and profitability of an investment in the host country; the third is the risk of ownership control, connected to events that influence the owner's ability to control and manage the investment (the investment could be expropriated or the initial owner could be forced to give a share of the property to the local partner at a lower price than expected).

It is easy to see how the assessment of CR can be somehow complex. Given the great multitude of definitions, there are different connotations and areas that can emerge as relevant. Studying this phenomenon is, therefore, complicated both for the methods and the variables to be used. One of the major criticisms of these studies, however, is the difficulty in forecasting any crises, because even the main measurement indicators of CR do not foresee, in the medium term, the changes or the vulnerability of the economies of the analyzed Countries (San-Martín-Albizuri and Rodríguez-Castellanos 2012).

First of all, it is useful to mention two important companies that measure CR in a different way: Euromoney and Coface. Euromoney evaluates the investment risk of a country, such as the risk of default on a bond, risk of losing direct investment, the risk to global business relations, etc., by taking a qualitative model, which seeks an expert opinion on risk variables within a country (90% weighting) and combining it with a basic quantitative value (10% weighting). The qualitative score is visible independently of the ECR score, and it reflects a snapshot of a country's current position. The ECR score is displayed on a 100-point scale, with 100 being nearly devoid of any risk, and 0 being completely exposed to every risk. Updated quarterly, the Coface Country Risk Assessment map offers a unique overview across 160 countries around the world. The Coface country risk assessment aims at evaluating the average credit risk of companies in a given country. The evaluation is based on economic, financial, and political data. However, it also takes into account Coface experience in the country, under two dimensions: Coface's payment experience on the companies of the country and also its assessment of the business climate. The most recent literature mainly focuses on expanding research through new methods and data. Brown et al. (2015) stress that new research can exploit "Big Data" which, as the name implies, offers the possibility to use a large amount of information, managed by means of more advanced methodologies and better suited to this new form of analysis: it is urgent to produce better models that can adapt to global change.

As stated by Cukier and Mayer-Schönberger (2013a), the concept of Big Data is linked to the transformation of our society and the increased use of the Internet. With the advent of the network, there is more and more information that can be shared and the

web has now become a real database. Indeed, it is estimated that only less than 2% of the data is still in a non-digital format. Using a large amount of data is one of the first changes that this approach brings. The data to be collected are many more than the small quantities or samples used by statisticians until the last century and are more useful because we can benefit from insights that would not be possible with less information. Unlike sampling, where the information obtained from sample statistics is extended to the entire population, Big Data gives us indications that are less precise overall, but more detailed in the subgroups, marking a transformation in the way in which a society processes information.

The analysis of this type of data is detached from the search for causality to give space to the relationship between the variables. It may not reveal why a particular phenomenon is happening but, based on the relationship between the data, it may prevent it. Thus, the concept of "Datafication" (Cukier and Mayer-Schönberger 2013b) was born, that is, the possibility to collect all kinds of data, even those that until recently could not be considered as such. The possibility of exploiting Big Data and its evolution in future years is also linked to the development of statistical systems because using a large amount of data involves the use of tools for collecting, organizing, storing, and analyzing more and more complex information compared to those used until now.

From the analysis of CR studies of the last decades, it appears that researchers have used different methods of quantitative analysis (the prevailing methodology) and developed evaluation models intending to identify the relationship between the CR indicators and the level of risk of countries (Kosmidou et al. 2008).

Among the most widespread quantitative methods used in the literature, there is the regression model (Agliardi et al. 2012), which uses the debt crisis as a dependent variable and, as independent variables, a number of economic, political, and institutional variables. The authors develop an approach for the construction of aggregate indexes for economic, political, and financial risks in emerging countries based on an analysis of the stochastic domain (SD). The approach has the advantage of providing an efficient index resulting from the less variable combination of risk factors. The model exploits a greater number of variables than other methods and one result of this research is that the type of risk that most affects the sovereign debt is found to be the financial risk. Regarding financial risk, two authors, Aboura and Chevallier (2015), propose an assessment of CR as a measure of the risk of financial markets, taking as the object of their study the US economy. They argue that CR derives from elements such as stocks, interest rates, or exchange rates between currencies.

The qualitative methodology is used by some authors such as Brown et al. (2015) as a complement to the index they propose, the Robinson Country-Risk Index (RCRI), focused on the multidimensionality of CR and the use of big data. They propose a dynamic index that incorporates four large dimensions: administration, economy, operations, and society. The qualitative assessments that accompany the proposed index try to tackle the complexity of the political, economic, and social aspects of risk without sacrificing the analysis of the context of the country under consideration. The mixed calculation of CR is also used in the analysis by Ivaldi (2013), who uses factorial analysis to build an index, the Factorial Country Risk Index (FCRI), which considers quantitative and qualitative variables.

The assessment of a country can be seen as a decision problem in which the decision-maker (for example an expert, a creditor, or an investor) tries to classify countries on the basis of numerous factors, i.e., the evaluation criteria (Cosset et al. 1992). In addition to multivariate analysis, there are authors (Cosset et al. 1992) who propose a multi-criteria method that aggregates the preferences of the decision-maker according to his/her criteria. The weights found suggest that the variables "Gross National Product Per Capita", "Propensity to Invest", "Current Account Balance on GNP", and "Export Variability" are the most important determinants of the solvency of a country.

In the wake of this model, ten years later, Doumpos and Zopounidis (2002) propose a new multi-criteria method called MHDIS (Multi-group Hierarchical DIScrimination),

intended as a non-parametric model for the development of a multi-class classification (International Rating Agencies classify countries in more than 3 classes). Hammer et al. (2004, 2006), in their studies, aim to develop transparent, consistent, self-sufficient, and stable Country Risk Assessment systems, in close relation with the CR ratings provided by Standard and Poor's Agency. They propose two models, the first one using the classical econometric technique of multiple linear regression and the second one using the logic-combinatorial technique, the LAD (Logical Analysis of Data) model. Both models are not recursive (i.e., they are not based on previous years' evaluations) and use economic, financial, and political variables. The two models were subsequently compared and the affinity in the results increases the reliability and validity of both the methods used. Finally, in the work by Hoti (2005), a multivariate econometric method (VARMA-GARCH) to analyze CR and spillover effects of a country is implemented. The author shows that, in the current state of world affairs, the economic and financial wealth and political power of a country are decisive elements for its dominant position in the international financial community and for its political status.

## 3. Methodology

In literature, a debate is still on for how it concerns the robustness of aggregative methods and their adaptability to the analysis of complex multidimensional systems (Maggino 2017). In recent years, non-aggregative analysis methods have become established as they allow us to overcome the limits, or part of them, that characterize aggregative methods (Alaimo et al. 2020; Ivaldi et al. 2020).

In general, the aggregative methods are characterized by the following strengths and weaknesses: most aggregative methods implicitly assigned the same weight to all the indicators; they tend to produce a compensatory effect unless a penalty is provided to offset the aggregation of particularly uneven values for a same statistical unit (Mazziotta and Pareto 2016); many aggregate indices are not suitable for longitudinal analysis.

Although like all statistical tools they have drawbacks, the field of aggregative methods continues to be frequently beaten by researchers. In this regard, many recent contributions have been developed on an aggregative analytical system (Ciacci et al. 2020; Mazziotta and Pareto 2020; Penco et al. 2020). Nowadays, thanks to their qualities, aggregative methods represent simple but useful tools for measuring and evaluating multidimensional systems. Aggregative methods are rather simple to construct and, at the same time, effective in pursuing communicative purposes. For this reason, in addition to their robustness, easy comprehensibility, and notoriety in the literature, we have chosen to measure CR through three different indexes of aggregative nature. Our choice fell on three aggregative methods, i.e., additive, weighted additive, and Peña Distance method. These methods, while presenting different characteristics, are adaptable to different research designs.

In this paragraph, we provide information about the variables' selection process (Section 3.1) and aggregative methods of analysis we have employed in the paper (Section 3.2 and the following subsection).

### 3.1. Variables Selections

From the analysis of the literature, we obtain 34 variables. The data concern the time period 2009–2016 and are extracted from the World Bank, the Worldwide Governance Indicators (WGI), and Eurostat databases for all 28 European Union countries. The following bulleted list shows the variables used and their polarity, taking into account that CR is measured in a positive direction (higher results identify better and less risky situations):

- Voice and Accountability (polarity +)
- Political Stability and Absence of Violence/Terrorism (+)
- Government Effectiveness (+)
- Regulatory Quality (+)
- Rule of Law (+)

- Control of Corruption (+)
- Imports of goods and services (% of GDP) (+)
- Total reserves (includes gold, current US$) (+)
- GDP growth (annual %) (+)
- Population growth (annual %) (+)
- Employment to population ratio, 15+, total (%) (modelled ILO estimate) (+)
- Population aged 15–64 (% of total) (+)
- Exports of goods and services (% of GDP) (+)
- General government deficit/surplus (% of GDP and million EUR) (−)
- GDP per capita (current US$) (+)
- Public debt (% of GDP) (−)
- Gini coefficient of equivalenced disposable income (−)
- Armed forces personnel (% of total labor force) (+)
- General government final consumption expenditure (annual % growth) (+)
- Health expenditure, total (% of GDP) (+)
- Inflation, consumer prices (annual %) (+)
- Life expectancy at birth, total (years) (+)
- Military expenditure (% of GDP) (+)
- Net external debt—annual data, % of GDP (−)
- Real effective exchange rate—Euro Area trading partners (+)
- Current account balance—annual data (% of GDP) (+)
- People at risk of poverty or social exclusion (−)
- Gross domestic expenditure on research and development (R&D) (+)
- Gender employment gap (−)
- Gender pay gap in unadjusted form (−)
- Population reporting occurrence of crime, violence, or vandalism in their area by poverty status (−)
- Investment by institutional sectors (+)
- Early leavers from education and training by sex (%) (−)
- Young people neither in employment nor in education and training by sex (%) (−)

The only missing data, the "net external debt % GDP" for the United Kingdom, is obtained by the average of the other (standardized) variables for the United Kingdom and reported by the inverse operation to the original scale value (Bannerjee et al. 2016).

To identify which variables among these better represent a CR index, we use the Principal Component Analysis (PCA) method, according to which, if two variables have a strong correlation with a dimension or factor not directly observable, a non-negligible part of the correlation between the two variables is explained by the fact that they have that dimension in common. For more references on the method, see References (Dillon and Goldstein 1984; Filmer and Pritchett 2001; Nardo et al. 2005; Gbetibouo et al. 2010; Barbanelli 2007; Pituch and Stevens 2016). It is recognized that the number of factors to be extracted is a choice that concerns both the safeguarding of the synthesis objective (the number of factors extracted must be lower than the original variables) and the guarantee of the adequacy of the solution in terms of the ability to reproduce the matrix of the eigenvectors R, i.e., the amount of variance that the transformation maintains. In this work, we use the Varimax rotation because it increases the simplicity of factors, that is, it maximizes the variance of the saturations of the variables within each factor. Therefore, it tends to make the higher saturations higher and the lower ones even lower, so that on each factor there are variables with high saturations and others with low saturations (Pituch and Stevens 2016). The criterion used for choosing the number of factors to consider was that the minimum variance explained overall by the factors of the solution be higher than 70% (Ivaldi et al. 2016), which in our case meant considering the first six components extracted. Therefore, 6 factors emerge from the PCA, on which the variables are distributed with an explained variance of over 70%, which implies that the variables arranged on the

last two factors are to be excluded, respectively, "Current account balance—annual data" and "Real effective exchange rate—Euro Area trading partners".

By this process, we obtain the weights to assign to each variable for all years (Table 1).

**Table 1.** Variables' weights for each year.

|  | 2009 | 2010 | 2011 | 2012 | 2013 | 2014 | 2015 | 2016 |
|---|---|---|---|---|---|---|---|---|
| Voice and Accountability | 0.292 | 0.285 | 0.284 | 0.289 | 0.286 | 0.280 | 0.281 | 0.278 |
| Political Stability and ... | 0.189 | 0.203 | 0.232 | 0.222 | 0.233 | 0.242 | 0.226 | 0.208 |
| Government Effectiveness | 0.286 | 0.278 | 0.276 | 0.275 | 0.266 | 0.272 | 0.272 | 0.282 |
| Regulatory Quality | 0.278 | 0.264 | 0.275 | 0.276 | 0.278 | 0.263 | 0.270 | 0.275 |
| Rule of Law | 0.288 | 0.278 | 0.280 | 0.282 | 0.279 | 0.281 | 0.285 | 0.288 |
| Control of Corruption | 0.293 | 0.286 | 0.282 | 0.279 | 0.273 | 0.276 | 0.277 | 0.282 |
| Imports of goods and service | 0.076 | 0.077 | 0.083 | 0.083 | 0.095 | 0.113 | 0.120 | 0.116 |
| Reserves | 0.054 | 0.054 | 0.041 | 0.047 | 0.048 | 0.041 | 0.038 | 0.029 |
| GDP growth (annual %) | 0.071 | 0.203 | 0.072 | 0.045 | 0.081 | 0.088 | 0.063 | 0.003 |
| Population growth (annual %) | 0.201 | 0.204 | 0.156 | 0.212 | 0.195 | 0.212 | 0.242 | 0.238 |
| Employment to population | 0.194 | 0.205 | 0.225 | 0.231 | 0.236 | 0.226 | 0.228 | 0.208 |
| Population ages 15–64 | 0.115 | 0.103 | 0.090 | 0.092 | 0.083 | 0.072 | 0.059 | 0.075 |
| Exports of goods and service | 0.113 | 0.111 | 0.113 | 0.107 | 0.114 | 0.130 | 0.137 | 0.129 |
| General government def surp | 0.140 | 0.058 | 0.150 | 0.093 | 0.124 | 0.147 | 0.150 | 0.092 |
| GDP per capita (current US$) | 0.260 | 0.256 | 0.251 | 0.253 | 0.250 | 0.250 | 0.253 | 0.259 |
| Public Debt | 0.051 | 0.031 | 0.001 | 0.004 | 0.013 | 0.032 | 0.039 | 0.030 |
| Gini coefficient | 0.157 | 0.178 | 0.203 | 0.192 | 0.180 | 0.178 | 0.171 | 0.157 |
| Armed forces personnel | 0.173 | 0.175 | 0.192 | 0.198 | 0.218 | 0.206 | 0.216 | 0.204 |
| General government final | 0.097 | 0.142 | 0.090 | 0.080 | 0.136 | 0.054 | 0.088 | 0.097 |
| Current health expenditure | 0.169 | 0.141 | 0.153 | 0.173 | 0.172 | 0.162 | 0.147 | 0.169 |
| Inflation, consumer prices | 0.077 | 0.070 | 0.107 | 0.112 | 0.017 | 0.023 | 0.018 | 0.087 |
| Life expectancy at birth | 0.214 | 0.200 | 0.176 | 0.177 | 0.147 | 0.158 | 0.165 | 0.173 |
| Military expenditure | 0.115 | 0.093 | 0.112 | 0.108 | 0.132 | 0.140 | 0.159 | 0.173 |
| Net external debt | 0.118 | 0.122 | 0.124 | 0.124 | 0.131 | 0.146 | 0.143 | 0.138 |
| Real effective exchange | 0.135 | 0.087 | 0.034 | 0.056 | 0.122 | 0.093 | 0.064 | 0.109 |
| Current account balance | 0.165 | 0.192 | 0.238 | 0.202 | 0.157 | 0.158 | 0.161 | 0.116 |
| People at risk of poverty | 0.018 | 0.035 | 0.016 | 0.068 | 0.011 | 0.129 | 0.003 | 0.035 |
| Gross domestic expenditure | 0.243 | 0.237 | 0.232 | 0.230 | 0.225 | 0.214 | 0.207 | 0.212 |
| Gender employment gap | 0.007 | 0.109 | 0.055 | 0.013 | 0.040 | 0.049 | 0.039 | 0.021 |
| Gender pay gap | 0.022 | 0.014 | 0.047 | 0.056 | 0.076 | 0.052 | 0.049 | 0.045 |
| Population | 0.072 | 0.027 | 0.058 | 0.057 | 0.044 | 0.060 | 0.075 | 0.079 |
| Investment by institutional | 0.129 | 0.060 | 0.008 | 0.026 | 0.042 | 0.045 | 0.071 | 0.126 |
| Early leavers from | 0.012 | 0.026 | 0.043 | 0.025 | 0.038 | 0.000 | 0.034 | 0.000 |
| Young people neither in | 0.212 | 0.235 | 0.242 | 0.249 | 0.257 | 0.250 | 0.242 | 0.249 |

### 3.2. Aggregative Methods

In order to define a measurement of CR that may have a validation criterion (Carr-Hill and Charmels-Dixon 2005), in this work we apply three different aggregative methods. The first one is a simple additive method (ADD), the second consists of a weighted additive index (WADD), while the third, Pena Distance (DP2), is a non-compensative method. This last is applied in the construction of composite indicators when the symmetry of variables or their weight is to be kept relevant and interpretable (Munda and Nardo 2009).

#### 3.2.1. Additive Method

Initially, the indexes are calculated in an additive way, adding up the contributions of the selected variables. Due to the lack of homogeneity of the selected variables, it is deemed necessary to standardize. This procedure avoids dependence on the units of measurement, which may affect data analysis (Han et al. 2012). Then, for each observation, the z-scores are calculated for each of the variables under examination, obtained by subtracting the value of the countries average at each observation and dividing the result by the average

square deviation of the countries. The single index, therefore, consists of the unweighted sum of three z-scores (Carstairs and Morris 1991; Townsend 1987; Forrest and Gordon 1993; Ivaldi and Testi 2011).

The CR index is calculated as the non-weighted sum of the $Z_i$:

$$Z_1 = \frac{x_1 - \mu x_1}{\sigma x_1} \; Z_2 = \frac{x_2 - \mu x_2}{\sigma x_2} \; \ldots \ldots Z_i = \frac{x_i - \mu x_i}{\sigma x_i} \ldots Z_n = \frac{x_n - \mu x_n}{\sigma x_n}$$

and being $\mu x_i$ and $\sigma x_i$ ($i = 34$) the means and the average square deviations of the variables under examination, the index results as:

$$\text{ADD} = -\sum_1^{34} Z_i$$

In order to make the index comparable over time, we "stack" the Countries for each year and then we calculate the index (Norman 2010; Landi et al. 2018).

$$\mu_j = \frac{\sum_{t=1}^{k} \sum_{i=1}^{n} x_{i,j,t}}{kn}$$

$$\sigma_j = \sqrt{\frac{\sum_{t=1}^{k} \sum_{i=1}^{n} (x_{i,j,t} - \mu_j)}{kn}}$$

where $i = 1, 2, \ldots, n$ are the number of countries, $j = 1, \ldots, m$ are the variables used in the indicator, and $t = 1, \ldots, k$ are the years of the data.

### 3.2.2. Peña Distance Method (DP2)

The third index follows a parametric non-compensative method that is applied to different sectors: economic and social cohesion (Molina et al. 2015), environmental quality (Montero et al. 2010), quality of life (Somarriba and Pena 2009; Somarriba and Pilar 2016), welfare systems (Martinez-Martinez et al. 2016), political participation (Ivaldi et al. 2017), deprivation (Ivaldi et al. 2018, and measure of objective and subjective health (Ivaldi et al. 2018).

The Peña Distance method (Pena 1977) allows solving the problems related to arbitrary weights by assigning weights to partial indicators on the basis of their correlation with the global index. The DP2 indicator also has many properties: non-negativity, commutability, triangular inequality, existence, determination, monotonicity, uniqueness, transitivity, invariance to change of origin and/or scale of the units in which the variables are defined, invariance to a change in the general conditions and exhaustiveness and reference base (Nayak and Mishra 2012). The DP2 indicator, providing the distance of each region from a reference base, which corresponds to the theoretical area achieving the lowest value of the variables being studied, is defined for area $j$ as follows:

$$DP2_j = \sum_{i=1}^{n} \left\{ \left( \frac{d_{ij}}{\sigma_i} \right) \left( 1 - R^2_{i,i-1,i-2,\ldots,1} \right) \right\}$$

With $i = 1, \ldots, n$; (areas) and $j = 1, 2, \ldots, m$ (variables)

$d_{ij} = \left| x_{ij} - x^*_{ij} \right|$ is the difference between the value taken by the $i$th variable in the area $j$ and the minimum of the variable in the least desirable theoretical scenario, namely, the reference value of the matrix X.

$\sigma_i$ is the standard deviation of variable $i$;

$R^2_{i,i-1,i-2,\ldots,1}$ is the coefficient of multiple linear correlations squared in the linear regression of $X_i$ over $X_{i-1}, X_{i-2}, \ldots, X_1$, and it indicates the part of the variance of $X_i$ explained linearly by the variables $X_{i-1}, X_{i-2}, \ldots, X_1$. This coefficient is an abstract number and it is unrelated to the measurement units of the different variables. So, $(1 - R^2_{i,i-1,i-2,\ldots,1})$ is the correction factor, which shows the variance part of $X_i$ not explained by the linear regression model.

## 4. Results and Discussion

Starting with the analysis of the correlation between the results obtained by applying the three methods, it emerges that Spearman's coefficient tends to be high in all the years (see Figure 1). In general, the high correlation between the different types of indexes demonstrates the robustness of the different measurements as well as their similarity. In particular, correction with the weights extracted from the factorial analysis tends to lead to similarity, if we look at WADD and DP2. More accentuated differences over time result between DP2 and ADD indexes. The correlation results also show the overall stability of the CR phenomenon over time. In other words, indexes not only show restrained deviations from one method to another but also differ very little in the results over a precise period of time. From this point of view, the weighted indexes are more robust, presenting higher correlation values over time than the unweighted additive. For the correlation matrix including all years, see Figure A1, Appendix A.

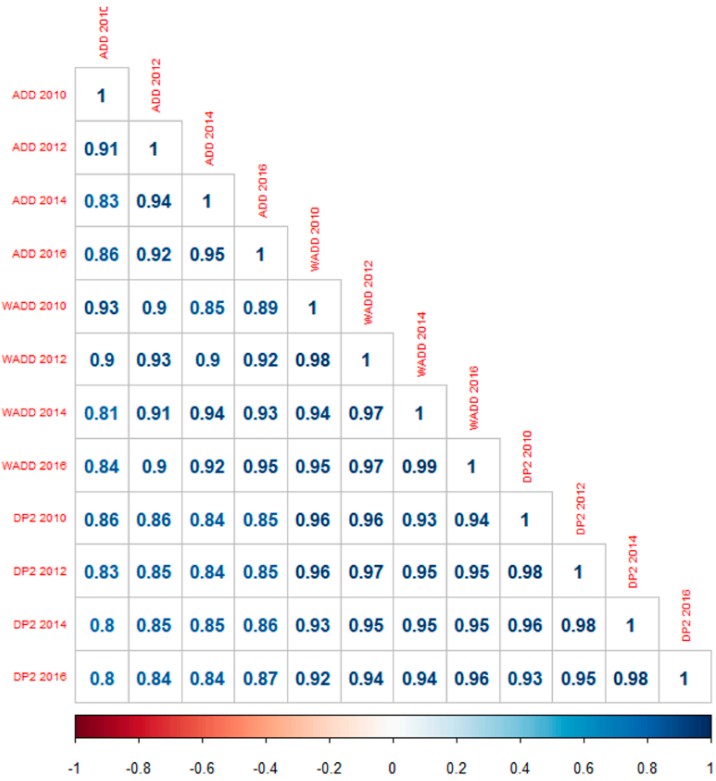

**Figure 1.** Spearman correlation matrix representation (years 2010, 2012, 2014, 2016).

The indexes assume values so that higher values correspond to better performance and, therefore, less CR. Analyzing the DP2 results in historical series (Figure 2), it is possible to notice that most European countries have constant CR values over time. This finding shows that, in the absence of shocks or special events, the assessment of the CR level tends to remain constant over a generally long period of time. However, there are exceptions, both in a negative and positive sense. Among the first cases are Greece, Hungary, and Poland, which show the highest increases in CR from 2009 to 2016. In particular, Hungary and Poland are burdened by political (in)stability factors (see Freedom 2016), while Greece is constantly burdened by economic factors. On the other hand, Croatia, Estonia, Romania, and Lithuania stand out positively. Estonia, Romania, and Lithuania were characterized by significant increases in GDP, indicating a process of consolidated economic growth over the years observed (https://ec.europa.eu/eurostat/databrowser/view/nama_10 _gdp/default/table?lang=en). It should be noted, however, that Croatia, despite the improvements, continues to settle at levels of CR far above average. Croatia, Estonia, and Romania also show a reduction in inequalities in wealth distribution, comparing

GINI coefficients from 2009 to 2016 (https://data.worldbank.org/indicator/SI.POV.GINI). Overall, there is a consolidation of perceived levels of democracy in these countries, with associated increases in government effectiveness, regulatory quality, political stability, rule of law, and freedom of expression. Low and constant levels of CR over time are observed in more developed countries, such as Austria, Belgium, Denmark, Finland, Germany, Sweden, and the United Kingdom. Bulgaria is a country that deviates from those just analyzed, due to a consistently high CR trend. Even the Balkan country increases its CR if we consider the whole period 2009–2016. Average CR levels, but with a recent positive trend, are found in Italy, Malta, Portugal, Cyprus, while a stable or negative average trend appears in France, the Czech Republic, Slovakia, and Spain. The French situation is burdened in particular by political instability and the threat of terrorist attacks, the high deficit, and the public debt situation, the latter two situations also common to Spain.

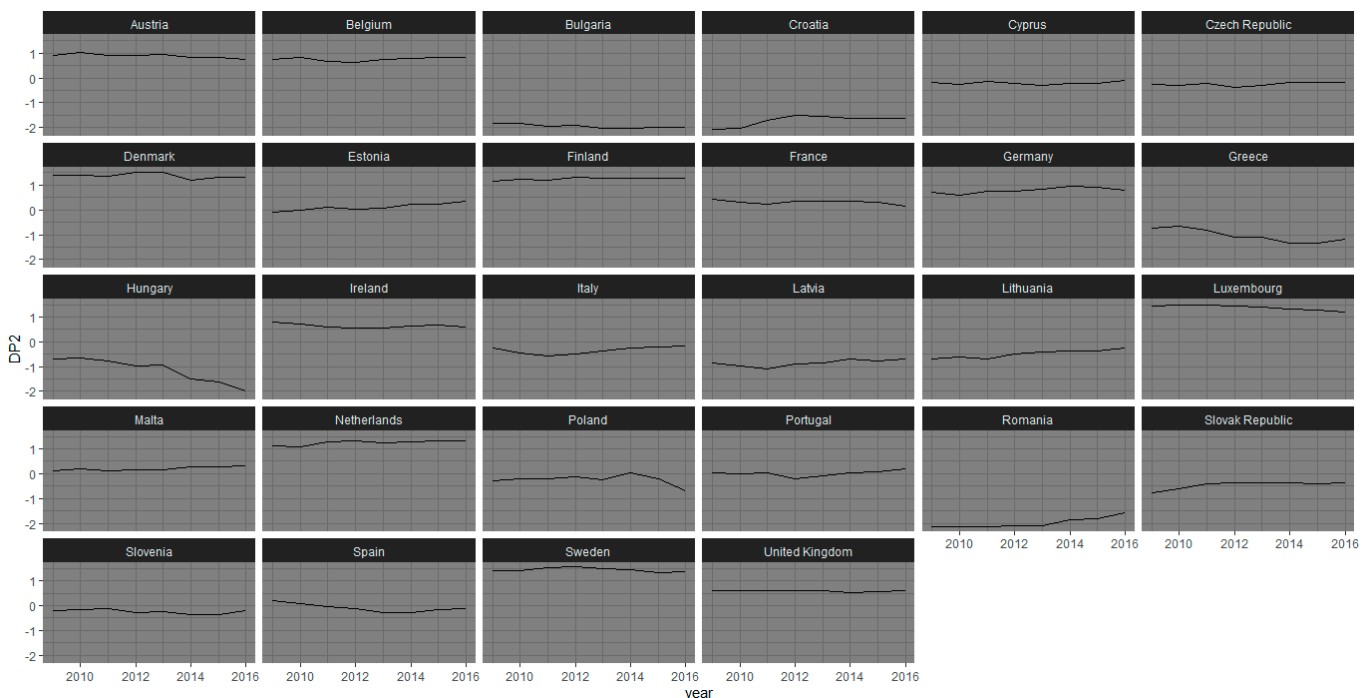

**Figure 2.** Historical series based on Pena Distance (DP$_2$) results.

Focusing on the results of the ADD (Figure 3), no macroscopic differences emerge compared to DP2, given the high correlation that characterizes the two indexes. However, it is possible to identify some countries with a slightly different CR trend from the previous one. The cases that stand out for the greatest differences are: Belgium, which has a more pronounced oscillatory trend and a more prolonged negative phase in this case; Cyprus which, contrary to the constant trend assumed previously, shows a sharp increase in CR in 2014, followed by a recovery phase; Estonia also shows a less linear development than the additive index, showing a clear decrease in risk in 2013. However, Finland has the most divergent measurements between DP2 and additive. In the case of additive, the Scandinavian country has a strongly negative phase from 2013 to 2015, only to see a reduction in the risk level. The cases of Latvia, Poland, and Sweden also show less constant trends over time, with more ripples. It is, therefore, clear that the unweighted additive index produces more abrupt changes in values than the weighted indexes that are more stable over time.

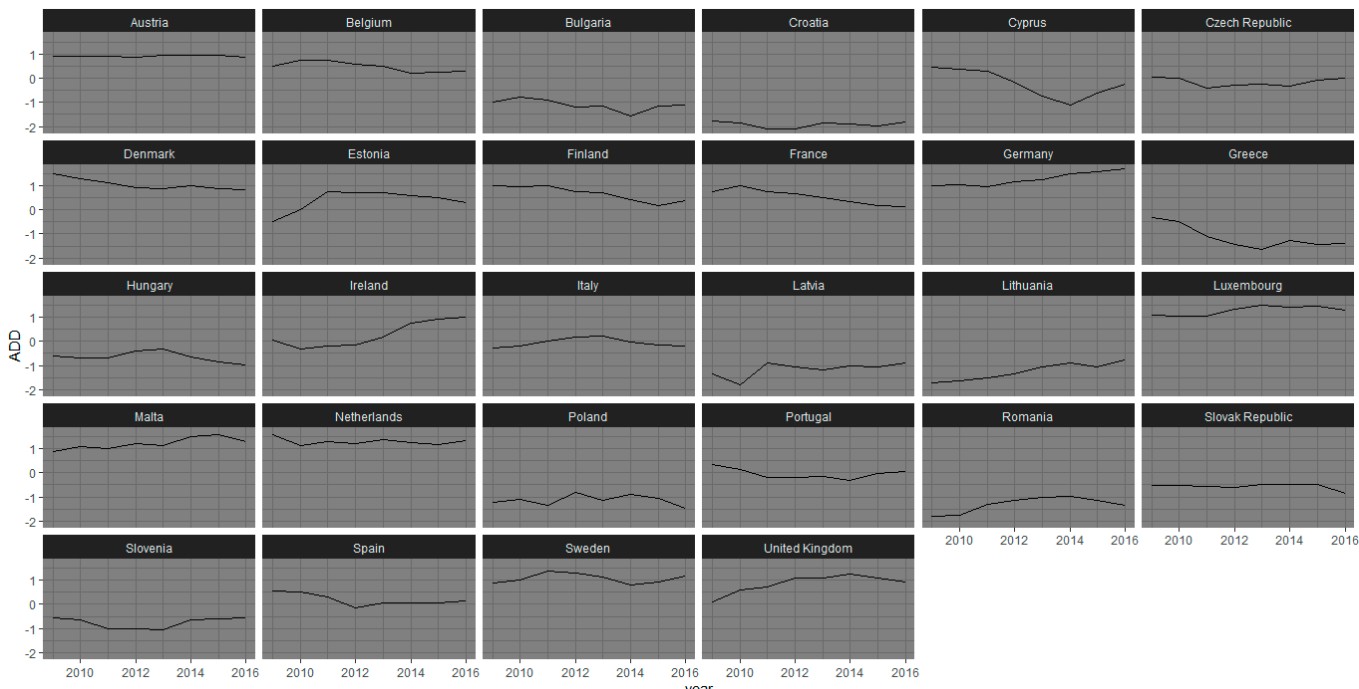

**Figure 3.** Historical series based on simple additive method (ADD) results.

Between WADD and DP2, both weighted indexes, the discrepancies are quite small. As previously written, the level of correlation between the results obtained by these methods is high. The greatest differences in the case of WADD (Figure 4) are found in Croatia, so it is possible to observe a less pronounced decrease in CR in 2012 than with the DP2 method. In the case of WADD aggregation method, the curve assumes a more linear trend over the entire period. The opposite situation in Cyprus, where in 2014, with WADD the CR increases while a linear trend is confirmed, with a positive trend in the case of DP2. With WADD measurement, in Finland, there is a marked increase in CR in 2015, while the decline in Hungary from 2013 onwards is less steep with WADD. For the remaining countries, there are little or no differences, compared with the results that emerged with the other methods of aggregation.

The proposed aggregation methods are robust, given the high correlation of the different distributions' results. In other words, the results tend to converge, although the proposed methods are characterized by differences in the calculation method. This conclusion allows us to state that the CR phenomenon can be measured by applying all the three proposed methods. The discriminant that allows you to apply one method rather than another is thus attributable to the researcher's choice to weigh or not the indicators that make up the index. Moreover, in this specific case, the effect of the weights extracted from PCA does not particularly influence the results.

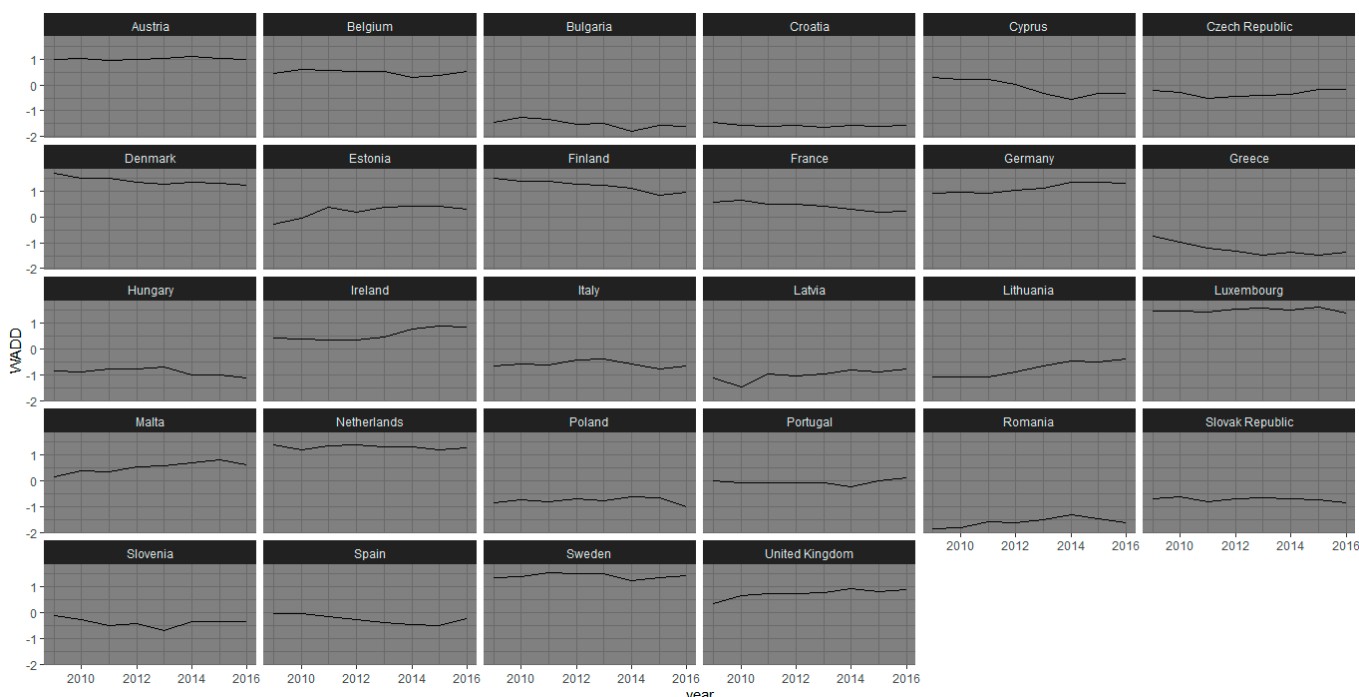

**Figure 4.** Historical series based on weighted additive index (WADD) results.

For the complete results including all years, see Figure A2, Appendix A.

## 5. Conclusions

In recent years, the deeper interconnection of financial markets may have influenced the level of instability of the countries' economies, whose performances do not depend only on their own choices and possibilities since increasingly more factors affect economic outcomes and trade relationships. Therefore, events like the recent real estate crisis in the US spread throughout the developed countries and affected their economies and the whole international trade. The CR indexes proposed to capture the fact that the economic recovery for many countries is still far and it depends not only on being a developed country but also on the capabilities of being an international trader able to face the challenges of globalization. In this perspective, where a context analysis cannot be ignored to make a valid evaluation of the economic performances, and in line with Meldrum's definition (Meldrum 2000), the present study proposes three measures of CR. In particular, as regarding the choice of the variables, the starting point is a large collection of variables already used in the past and available for European countries. The second step is the selection of the variables using factor analysis. Then, three indexes are constructed to measure CR for each country. The aim of the study is to present operational and easily computable methods to measure CR. The dissemination of these measurement methods can allow economic and political actors to apply effective and easily communicable solutions at the same time. The different methods applied despite the fact that two indexes are weighed and the additive is not, give very similar results and this suggests the robustness of the measures. This finding allows us to state CR can be measured by applying all the three proposed methods. A researcher can apply one of the three methods here proposed, thinking about the need to weigh the indicators that make up the index. Among the proposed methods, however, there are some differences related to the adoption of a certain methodological framework. In the case in which all the indicators that make up the index are considered of equal importance in defining the construct, then the unweighted solution is chosen. In this case, it is represented by the additive index. On the other hand, when it is found the need to tie the construction of the index to the assignment of weights, it is possible to converge toward the solution offered by the weighted cognitive index or DP2. In the latter case,

the factorial analysis preparatory to the definition of weights is an integral and not an additional step in the procedure for calculating the index. The choice to build an index among those proposed to assess the level of CR is due to the following reasons: the ease of calculation, the simplicity in the interpretation of the results, the comparability to a given reference year and over time.

These measures of CR present many limitations as far as they are static representations of dynamic characteristics of evolving phenomena, such as the economy of a country and its external relationships with other agents. In the next few years, the growing availability of Big Data will pose new challenges and opportunities in estimating CR, enabling more accurate measures in the context of increasing globalization.

**Author Contributions:** Study conceptualization, G.B., A.C., E.I.; Methodology, G.B. and A.C.; data collection, G.B. Supervision, E.I.; and all authors have written and revised the text of this paper. All authors have read and agreed to the published version of the manuscript.

**Funding:** This research received no external funding.

**Institutional Review Board Statement:** Not applicable.

**Informed Consent Statement:** Not applicable.

**Data Availability Statement:** Not applicable.

**Conflicts of Interest:** The authors declare no conflict of interest.

## Appendix A

| | ADD 2009 | ADD 2010 | ADD 2011 | ADD 2012 | ADD 2013 | ADD 2014 | ADD 2015 | ADD 2016 | WAD D 2009 | WAD D 2010 | WAD D 2011 | WAD D 2012 | WAD D 2013 | WAD D 2014 | WAD D 2015 | WAD D 2016 | DP2 2009 | DP2 2010 | DP2 2011 | DP2 2012 | DP2 2013 | DP2 2014 | DP2 2015 | DP2 2016 |
|---|---|---|---|---|---|---|---|---|---|---|---|---|---|---|---|---|---|---|---|---|---|---|---|---|
| ADD 2009 | 1 | 0.97 | 0.91 | 0.86 | 0.83 | 0.8 | 0.83 | 0.85 | 0.93 | 0.93 | 0.9 | 0.89 | 0.86 | 0.81 | 0.82 | 0.84 | 0.88 | 0.87 | 0.87 | 0.84 | 0.84 | 0.81 | 0.82 | 0.81 |
| ADD 2010 | 0.97 | 1 | 0.95 | 0.91 | 0.88 | 0.83 | 0.85 | 0.86 | 0.9 | 0.93 | 0.9 | 0.9 | 0.87 | 0.81 | 0.83 | 0.84 | 0.86 | 0.86 | 0.85 | 0.83 | 0.82 | 0.8 | 0.81 | 0.8 |
| ADD 2011 | 0.91 | 0.95 | 1 | 0.97 | 0.94 | 0.88 | 0.9 | 0.9 | 0.88 | 0.92 | 0.94 | 0.93 | 0.92 | 0.87 | 0.87 | 0.88 | 0.86 | 0.85 | 0.85 | 0.84 | 0.84 | 0.84 | 0.85 | 0.85 |
| ADD 2012 | 0.86 | 0.91 | 0.97 | 1 | 0.98 | 0.94 | 0.94 | 0.92 | 0.85 | 0.9 | 0.92 | 0.93 | 0.94 | 0.91 | 0.91 | 0.9 | 0.86 | 0.86 | 0.86 | 0.85 | 0.86 | 0.85 | 0.86 | 0.84 |
| ADD 2013 | 0.83 | 0.88 | 0.94 | 0.98 | 1 | 0.97 | 0.96 | 0.94 | 0.83 | 0.88 | 0.91 | 0.92 | 0.94 | 0.91 | 0.91 | 0.91 | 0.85 | 0.84 | 0.85 | 0.84 | 0.85 | 0.84 | 0.85 | 0.84 |
| ADD 2014 | 0.8 | 0.83 | 0.88 | 0.94 | 0.97 | 1 | 0.98 | 0.95 | 0.81 | 0.85 | 0.87 | 0.9 | 0.92 | 0.94 | 0.93 | 0.92 | 0.85 | 0.84 | 0.84 | 0.84 | 0.85 | 0.85 | 0.86 | 0.84 |
| ADD 2015 | 0.83 | 0.85 | 0.9 | 0.94 | 0.96 | 0.98 | 1 | 0.98 | 0.82 | 0.87 | 0.88 | 0.9 | 0.92 | 0.93 | 0.94 | 0.92 | 0.84 | 0.84 | 0.84 | 0.83 | 0.84 | 0.84 | 0.85 | 0.84 |
| ADD 2016 | 0.85 | 0.86 | 0.9 | 0.92 | 0.94 | 0.95 | 0.98 | 1 | 0.86 | 0.89 | 0.9 | 0.92 | 0.93 | 0.93 | 0.95 | 0.95 | 0.86 | 0.85 | 0.85 | 0.85 | 0.85 | 0.86 | 0.87 | 0.87 |
| WAD D2009 | 0.93 | 0.9 | 0.88 | 0.85 | 0.83 | 0.81 | 0.82 | 0.86 | 1 | 0.99 | 0.97 | 0.97 | 0.94 | 0.92 | 0.92 | 0.93 | 0.96 | 0.96 | 0.96 | 0.96 | 0.95 | 0.92 | 0.93 | 0.91 |
| WAD D2010 | 0.93 | 0.93 | 0.92 | 0.9 | 0.88 | 0.85 | 0.87 | 0.89 | 0.99 | 1 | 0.98 | 0.98 | 0.96 | 0.94 | 0.94 | 0.95 | 0.96 | 0.96 | 0.96 | 0.96 | 0.95 | 0.93 | 0.94 | 0.92 |
| WAD D2011 | 0.9 | 0.9 | 0.94 | 0.92 | 0.91 | 0.87 | 0.88 | 0.9 | 0.97 | 0.98 | 1 | 0.99 | 0.98 | 0.95 | 0.96 | 0.96 | 0.95 | 0.95 | 0.96 | 0.96 | 0.95 | 0.94 | 0.94 | 0.93 |
| WAD D2012 | 0.89 | 0.9 | 0.93 | 0.93 | 0.92 | 0.9 | 0.9 | 0.92 | 0.97 | 0.98 | 0.99 | 1 | 0.99 | 0.97 | 0.97 | 0.97 | 0.96 | 0.96 | 0.97 | 0.97 | 0.97 | 0.95 | 0.96 | 0.94 |
| WAD D2013 | 0.86 | 0.87 | 0.92 | 0.94 | 0.94 | 0.92 | 0.92 | 0.93 | 0.94 | 0.96 | 0.98 | 0.99 | 1 | 0.98 | 0.98 | 0.98 | 0.94 | 0.95 | 0.95 | 0.96 | 0.96 | 0.95 | 0.95 | 0.94 |
| WAD D2014 | 0.81 | 0.81 | 0.87 | 0.91 | 0.91 | 0.94 | 0.93 | 0.93 | 0.92 | 0.94 | 0.95 | 0.97 | 0.98 | 1 | 0.99 | 0.99 | 0.93 | 0.93 | 0.94 | 0.95 | 0.96 | 0.95 | 0.96 | 0.94 |
| WAD D2015 | 0.82 | 0.83 | 0.87 | 0.91 | 0.91 | 0.93 | 0.94 | 0.95 | 0.92 | 0.94 | 0.96 | 0.97 | 0.98 | 0.99 | 1 | 0.99 | 0.93 | 0.93 | 0.94 | 0.95 | 0.95 | 0.95 | 0.95 | 0.94 |
| WAD D2016 | 0.84 | 0.84 | 0.88 | 0.9 | 0.91 | 0.92 | 0.92 | 0.95 | 0.93 | 0.95 | 0.96 | 0.97 | 0.98 | 0.99 | 0.99 | 1 | 0.94 | 0.94 | 0.95 | 0.95 | 0.96 | 0.95 | 0.96 | 0.96 |
| DP2 2009 | 0.88 | 0.86 | 0.86 | 0.86 | 0.85 | 0.85 | 0.84 | 0.86 | 0.96 | 0.96 | 0.95 | 0.96 | 0.94 | 0.93 | 0.93 | 0.94 | 1 | 1 | 0.99 | 0.98 | 0.98 | 0.96 | 0.96 | 0.93 |
| DP2 2010 | 0.87 | 0.86 | 0.85 | 0.86 | 0.84 | 0.84 | 0.84 | 0.85 | 0.96 | 0.96 | 0.95 | 0.96 | 0.95 | 0.93 | 0.93 | 0.94 | 1 | 1 | 0.99 | 0.98 | 0.98 | 0.96 | 0.96 | 0.93 |
| DP2 2011 | 0.87 | 0.85 | 0.85 | 0.86 | 0.85 | 0.84 | 0.84 | 0.85 | 0.96 | 0.96 | 0.96 | 0.97 | 0.95 | 0.94 | 0.94 | 0.95 | 0.99 | 0.99 | 1 | 0.99 | 0.99 | 0.97 | 0.97 | 0.94 |
| DP2 2012 | 0.84 | 0.83 | 0.84 | 0.85 | 0.84 | 0.84 | 0.83 | 0.85 | 0.96 | 0.96 | 0.96 | 0.97 | 0.96 | 0.95 | 0.95 | 0.95 | 0.98 | 0.98 | 0.99 | 1 | 1 | 0.98 | 0.98 | 0.95 |
| DP2 2013 | 0.84 | 0.82 | 0.84 | 0.86 | 0.85 | 0.85 | 0.84 | 0.85 | 0.95 | 0.95 | 0.95 | 0.97 | 0.96 | 0.96 | 0.95 | 0.96 | 0.98 | 0.98 | 0.99 | 1 | 1 | 0.98 | 0.98 | 0.96 |
| DP2 2014 | 0.81 | 0.8 | 0.84 | 0.85 | 0.84 | 0.85 | 0.84 | 0.86 | 0.92 | 0.93 | 0.94 | 0.95 | 0.95 | 0.95 | 0.95 | 0.95 | 0.96 | 0.96 | 0.97 | 0.98 | 0.98 | 1 | 1 | 0.98 |
| DP2 2015 | 0.82 | 0.81 | 0.85 | 0.86 | 0.85 | 0.86 | 0.85 | 0.87 | 0.93 | 0.94 | 0.94 | 0.96 | 0.95 | 0.96 | 0.95 | 0.96 | 0.96 | 0.96 | 0.97 | 0.98 | 0.98 | 1 | 1 | 0.99 |
| DP2 2016 | 0.81 | 0.8 | 0.85 | 0.84 | 0.84 | 0.84 | 0.84 | 0.87 | 0.91 | 0.92 | 0.93 | 0.94 | 0.94 | 0.94 | 0.94 | 0.96 | 0.93 | 0.93 | 0.94 | 0.95 | 0.96 | 0.98 | 0.99 | 1 |

**Figure A1.** Appendix: Spearman correlation matrix.

| Country | ADD 2009 | ADD 2010 | ADD 2011 | ADD 2012 | ADD 2013 | ADD 2014 | ADD 2015 | ADD 2016 | WADD 2009 | WADD 2010 | WADD 2011 | WADD 2012 | WADD 2013 | WADD 2014 | WADD 2015 | WADD 2016 | DP2 2009 | DP2 2010 | DP2 2011 | DP2 2012 | DP2 2013 | DP2 2014 | DP2 2015 | DP2 2016 |
|---|---|---|---|---|---|---|---|---|---|---|---|---|---|---|---|---|---|---|---|---|---|---|---|---|
| Austria | 0.91 | 0.89 | 0.89 | 0.87 | 0.94 | 0.97 | 0.94 | 0.88 | 1.00 | 1.01 | 0.95 | 1.00 | 1.04 | 1.11 | 1.02 | 1.00 | 0.91 | 1.04 | 0.90 | 0.91 | 0.95 | 0.84 | 0.81 | 0.73 |
| Belgium | 0.49 | 0.73 | 0.73 | 0.57 | 0.51 | 0.19 | 0.25 | 0.29 | 0.45 | 0.60 | 0.57 | 0.53 | 0.53 | 0.28 | 0.36 | 0.54 | 0.76 | 0.82 | 0.67 | 0.64 | 0.73 | 0.77 | 0.82 | 0.84 |
| Bulgaria | -0.99 | -0.79 | -0.90 | -1.21 | -1.17 | -1.55 | -1.15 | -1.09 | -1.46 | -1.27 | -1.34 | -1.52 | -1.47 | -1.78 | -1.56 | -1.60 | -1.82 | -1.83 | -1.93 | -1.89 | -2.01 | -2.04 | -1.97 | -1.97 |
| Croatia | -1.78 | -1.86 | -2.10 | -2.08 | -1.86 | -1.88 | -1.97 | -1.81 | -1.46 | -1.55 | -1.60 | -1.58 | -1.64 | -1.57 | -1.59 | -1.58 | -2.06 | -2.01 | -1.72 | -1.51 | -1.54 | -1.62 | -1.61 | -1.62 |
| Cyprus | 0.46 | 0.38 | 0.28 | -0.17 | -0.73 | -1.10 | -0.62 | -0.24 | 0.31 | 0.23 | 0.20 | 0.02 | -0.34 | -0.54 | -0.32 | -0.32 | -0.16 | -0.26 | -0.14 | -0.23 | -0.31 | -0.20 | -0.22 | -0.11 |
| Czech Republic | 0.05 | 0.02 | -0.39 | -0.28 | -0.25 | -0.34 | -0.08 | 0.00 | -0.19 | -0.27 | -0.50 | -0.46 | -0.39 | -0.38 | -0.17 | -0.18 | -0.27 | -0.29 | -0.23 | -0.37 | -0.30 | -0.19 | -0.19 | -0.17 |
| Denmark | 1.50 | 1.27 | 1.12 | 0.93 | 0.87 | 0.99 | 0.86 | 0.84 | 1.68 | 1.50 | 1.50 | 1.35 | 1.28 | 1.33 | 1.30 | 1.21 | 1.38 | 1.39 | 1.34 | 1.51 | 1.51 | 1.21 | 1.31 | 1.32 |
| Estonia | -0.50 | -0.01 | 0.75 | 0.72 | 0.69 | 0.57 | 0.52 | 0.30 | -0.30 | -0.04 | 0.39 | 0.19 | 0.37 | 0.40 | 0.40 | 0.31 | -0.10 | 0.00 | 0.10 | 0.01 | 0.05 | 0.22 | 0.24 | 0.36 |
| Finland | 0.99 | 0.94 | 0.99 | 0.76 | 0.69 | 0.43 | 0.19 | 0.36 | 1.48 | 1.37 | 1.36 | 1.27 | 1.21 | 1.10 | 0.83 | 0.97 | 1.14 | 1.22 | 1.21 | 1.31 | 1.26 | 1.26 | 1.28 | 1.29 |
| France | 0.76 | 0.98 | 0.76 | 0.66 | 0.49 | 0.35 | 0.17 | 0.12 | 0.58 | 0.66 | 0.48 | 0.48 | 0.42 | 0.28 | 0.17 | 0.21 | 0.42 | 0.31 | 0.21 | 0.34 | 0.33 | 0.35 | 0.29 | 0.15 |
| Germany | 1.01 | 1.02 | 0.95 | 1.18 | 1.24 | 1.51 | 1.57 | 1.69 | 0.90 | 0.96 | 0.93 | 1.04 | 1.09 | 1.34 | 1.33 | 1.31 | 0.72 | 0.61 | 0.74 | 0.76 | 0.82 | 0.96 | 0.92 | 0.81 |
| Greece | -0.31 | -0.47 | -1.11 | -1.45 | -1.64 | -1.26 | -1.44 | -1.39 | -0.76 | -0.96 | -1.21 | -1.32 | -1.47 | -1.38 | -1.48 | -1.36 | -0.74 | -0.66 | -0.83 | -1.09 | -1.08 | -1.33 | -1.33 | -1.20 |
| Hungary | -0.61 | -0.68 | -0.68 | -0.42 | -0.31 | -0.65 | -0.84 | -0.98 | -0.84 | -0.89 | -0.77 | -0.78 | -0.69 | -1.03 | -1.02 | -1.14 | -0.68 | -0.67 | -0.78 | -0.96 | -0.95 | -1.51 | -1.60 | -1.97 |
| Ireland | 0.06 | -0.33 | -0.21 | -0.15 | 0.17 | 0.76 | 0.93 | 0.98 | 0.42 | 0.39 | 0.32 | 0.33 | 0.44 | 0.78 | 0.88 | 0.83 | 0.78 | 0.70 | 0.61 | 0.57 | 0.57 | 0.63 | 0.67 | 0.59 |
| Italy | -0.26 | -0.18 | 0.01 | 0.19 | 0.21 | -0.02 | -0.17 | -0.20 | -0.66 | -0.60 | -0.64 | -0.45 | -0.40 | -0.60 | -0.79 | -0.65 | -0.27 | -0.45 | -0.56 | -0.51 | -0.38 | -0.26 | -0.21 | -0.15 |
| Latvia | -1.37 | -1.80 | -0.88 | -1.07 | -1.20 | -1.01 | -1.07 | -0.91 | -1.14 | -1.48 | -0.98 | -1.07 | -0.98 | -0.83 | -0.91 | -0.80 | -0.87 | -0.99 | -1.08 | -0.89 | -0.86 | -0.68 | -0.76 | -0.70 |
| Lithuania | -1.74 | -1.64 | -1.52 | -1.33 | -1.05 | -0.89 | -1.07 | -0.78 | -1.08 | -1.10 | -1.10 | -0.90 | -0.68 | -0.47 | -0.52 | -0.38 | -0.70 | -0.61 | -0.71 | -0.48 | -0.41 | -0.37 | -0.39 | -0.25 |
| Luxembourg | 1.08 | 1.06 | 1.06 | 1.35 | 1.48 | 1.42 | 1.44 | 1.27 | 1.47 | 1.45 | 1.43 | 1.53 | 1.58 | 1.49 | 1.61 | 1.38 | 1.44 | 1.50 | 1.50 | 1.44 | 1.42 | 1.30 | 1.29 | 1.20 |
| Malta | 0.86 | 1.10 | 0.99 | 1.23 | 1.12 | 1.48 | 1.58 | 1.29 | 0.14 | 0.37 | 0.34 | 0.52 | 0.57 | 0.70 | 0.80 | 0.60 | 0.12 | 0.20 | 0.13 | 0.15 | 0.16 | 0.26 | 0.28 | 0.32 |
| Netherlands | 1.60 | 1.13 | 1.29 | 1.23 | 1.37 | 1.25 | 1.16 | 1.34 | 1.39 | 1.18 | 1.36 | 1.38 | 1.32 | 1.30 | 1.21 | 1.28 | 1.14 | 1.10 | 1.30 | 1.32 | 1.25 | 1.27 | 1.33 | 1.31 |
| Poland | -1.22 | -1.11 | -1.35 | -0.82 | -1.13 | -0.89 | -1.04 | -1.49 | -0.87 | -0.75 | -0.81 | -0.69 | -0.79 | -0.61 | -0.66 | -1.01 | -0.29 | -0.22 | -0.21 | -0.12 | -0.26 | 0.03 | -0.19 | -0.70 |
| Portugal | 0.36 | 0.14 | -0.18 | -0.21 | -0.17 | -0.31 | -0.04 | 0.06 | -0.01 | -0.07 | -0.08 | -0.07 | -0.07 | -0.23 | 0.00 | 0.12 | 0.04 | 0.01 | 0.03 | -0.21 | -0.08 | 0.04 | 0.07 | 0.21 |
| Romania | -1.82 | -1.74 | -1.32 | -1.12 | -1.03 | -0.97 | -1.14 | -1.36 | -1.87 | -1.83 | -1.59 | -1.64 | -1.51 | -1.31 | -1.46 | -1.62 | -2.13 | -2.15 | -2.15 | -2.11 | -2.08 | -1.86 | -1.81 | -1.57 |
| Slovak Republic | -0.51 | -0.51 | -0.56 | -0.59 | -0.48 | -0.50 | -0.50 | -0.87 | -0.69 | -0.62 | -0.82 | -0.71 | -0.68 | -0.70 | -0.73 | -0.86 | -0.79 | -0.62 | -0.40 | -0.36 | -0.36 | -0.37 | -0.41 | -0.36 |
| Slovenia | -0.54 | -0.63 | -1.00 | -1.00 | -1.07 | -0.63 | -0.59 | -0.56 | -0.11 | -0.29 | -0.49 | -0.43 | -0.69 | -0.35 | -0.36 | -0.33 | -0.20 | -0.18 | -0.13 | -0.29 | -0.25 | -0.38 | -0.35 | -0.22 |
| Spain | 0.54 | 0.51 | 0.29 | -0.14 | 0.07 | 0.04 | 0.06 | 0.14 | -0.05 | -0.04 | -0.15 | -0.26 | -0.37 | -0.48 | -0.50 | -0.24 | 0.20 | 0.06 | -0.04 | -0.12 | -0.28 | -0.28 | -0.18 | -0.13 |
| Sweden | 0.87 | 0.99 | 1.37 | 1.28 | 1.14 | 0.79 | 0.93 | 1.17 | 1.34 | 1.39 | 1.55 | 1.51 | 1.52 | 1.22 | 1.35 | 1.43 | 1.41 | 1.40 | 1.53 | 1.55 | 1.49 | 1.45 | 1.34 | 1.38 |
| United Kingdom | 0.11 | 0.61 | 0.71 | 1.09 | 1.10 | 1.26 | 1.10 | 0.94 | 0.33 | 0.66 | 0.72 | 0.73 | 0.79 | 0.94 | 0.81 | 0.88 | 0.61 | 0.59 | 0.59 | 0.61 | 0.62 | 0.51 | 0.56 | 0.61 |

**Figure A2.** Appendix: indexes results matrix.

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
