# Peer review of "Different Measures of Country Risk: An Application to European Countries"

_jrfm, doi:10.3390/jrfm14010019_

Round 1

Reviewer 1 Report

This article develops indexes for the European Union countries by applying three different methods in the field of formative approach.

I really like this paper well structured, clearly written, and presented in a logical and useful manner.

The title and the abstract are understandable and adequate. The references provided are adequate and up to date and the methodology used is clearly outlined. The figures and the tables are properly shown.
The analysis of the literature is well conducted and updated.

Discussion of results and the concluding remarks are clear and organic. I think it would be interesting to extend the research to other regions using the same criteria in a new paper.

The authors stated in the abstract "Then we proposed three simple aggregative processes in order  to obtain CR measures, at a precise time and over time."

The second point, the measure over time is really an interesting point, but I believe is not presented clearly in the paper how they addressed the issue. 

I invite the author to explain better this part. 

Reviewer 2 Report

The study of the results of the application of different methods for determining country risk is important for the orientation of the users due to the large number of institutions publishing the results of their own research. In this respect, the paper submitted for review is of interest to a wide range of stakeholders.

The following remarks may be made on the paper:

The formulation of the research goal in the abstract (“The aim is to provide suggestions on how CR can be measured”) does not fully correspond to the goal set in the introduction (“The purpose of the work is to build robust and easily updatable CR indexes”) and in fact, it does not accurately reflect the research idea of ​​the paper and its implementation.

The literature review tries to present the broad concept of "country risk" in a very concise format. The authors could expand the review by discussing some of the most popular methodologies used, for example, by Euromoney and the largest credit rating agencies.

Paragraph 2 “Methodology” states that “non-aggregative analysis methods have become established as they allow us to overcome the limits, or part of them, that characterize aggregative methods”. Would you clarify the limitations of aggregative methods and justify your choice of methodologies used.

Probably a technical omission, but paragraph 3.2. and subparagraph 3.2.1. have the same titles.

The results of the research and the recommendations to the researchers could be formulated more convincingly. Since no significant deviations in the results of the applied research methods are registered, then let substantiated recommendations be given under what conditions and under what set goals it is better to use one or another method.
